

# Aerosol-cloud interactions in liquid-phase clouds under different meteorological and aerosol backgrounds

Jianqi Zhao[1][2], Xiaoyan Ma[1] and Johannes Quaas[2]

[1]China Meteorological Administration Aerosol-Cloud and Precipitation Key Laboratory, Nanjing University of Information Science and Technology, Nanjing 210044, China

[2]Leipzig Institute for Meteorology, Leipzig University, Leipzig, Germany

Correspondence: Xiaoyan Ma (xma@nuist.edu.cn)

**Abstract.** We conduct a comparative analysis of aerosol-cloud responses in liquid-phase clouds under different aerosol and meteorological conditions based on simulations using the WRF-Chem-SBM model to improve our understanding of aerosol-cloud interactions. This study reveals that in relatively unstable but dry atmosphere, aerosols uplift cloud top height but have no significant impact on cloud thickness, while also suppressing precipitation in clean conditions (sea salt aerosol only). In relatively stable but humid atmosphere, aerosols significantly increase both cloud top height and cloud thickness. Although aerosols also suppress precipitation in clean conditions, they promote the occurrence of relatively intense precipitation by facilitating the development of deep clouds. Aerosols have both enhancing and weakening effects on cloud liquid water path (CLWP). The weakening occurs mainly through two mechanisms: 1) by increasing $N_d$ in thin clouds within a dry atmosphere, leading to smaller droplet sizes, which enhances evaporation within clouds and thus reduces CLWP. 2) By lifting cloud top height, facilitating the transition of liquid-phase clouds into mixed-phase or ice-phase clouds. The enhancing effect becomes more pronounced in environments with a relatively high column-averaged relative humidity, and is also modulated by atmospheric stability: 1) under low lower tropospheric stability (LTS), aerosols cause a relatively brief, explosive increase in CLWP. 2) Under high LTS, aerosols lead to relatively persistent increase in CLWP. For the liquid-phase clouds in the study, aerosols affect cloud development but have no significant impact on cloud lifetime, and precipitation affects the short-term variation of $N_d$ but does not change its overall trend.

## 1 Introduction

Clouds play a crucial role in the Earth radiation budget (Andersen et al., 2017). Aerosols can impose significant impacts on the microphysical properties, amount, and lifetime of clouds by serving as cloud condensation nuclei (CCN, the explanations of all the acronyms in the text can be found in Table S1) and ice nucleating particles and by influencing radiation (Twomey, 1977; Albrecht, 1989; Schultze and Rockel, 2018; Jia et al., 2019a). The former, also known as aerosol-cloud interactions, remain one of the major sources of uncertainty in climate assessment and prediction (IPCC, 2023) due to the complexity of the influencing factors (Jia et al., 2019b) and the difficulty in assessing these (Lohmann and Feichter, 2005; McComiskey and Feingold, 2012).

Liquid-phase clouds are the main contributors to the total impact of aerosols on clouds due to their broad coverage, high optical thickness, and often long lifetime (Chen et al., 2011; Andersen et al., 2017; Jia et al., 2019c). In addition, this type of cloud is highly sensitive to aerosol (Chen et al., 2014; Jia et al., 2019c). In liquid-phase clouds, aerosol-cloud interactions are primarily based on two mechanisms: 1) the Twomey effect, i.e. the entry of additional aerosol into the cloud leads to an increase in the cloud droplet number concentration ($N_d$), a decrease in the cloud droplet effective radius (CER), and an increase in cloud albedo under constant cloud water content (Twomey, 1977). 2) Rapid adjustments, consist of the response of cloud liquid water path (CLWP) and cloud fraction to changes in $N_d$ via the Twomey effect (Albrecht, 1989; Haghighatnasab et al., 2022; Jia et al., 2022). The responses of both to perturbations of $N_d$ vary as a



function of meteorological conditions, while the variation of $N_d$ with aerosol number concentrations ($N_a$) also differs under different aerosol conditions. The study by Zhao et al. (2024) indicated that at low $N_a$, $N_d$ increases with $N_a$, but at high $N_a$, an increase in $N_a$ may also lead to a decrease in $N_d$. Some statistical analyses of satellite-retrieved data suggested that an increase in aerosols, which leads to an increase in $N_d$, results in a decrease in marine low cloud CLWP (Michibata

et al., 2016; Rosenfeld et al., 2019). The study by Gryspeerdt et al. (2019) found that this decrease occurs in clouds with low precipitation probability and high $N_d$, while in clouds with low $N_d$ and droplet sizes above the precipitation threshold, aerosol-induced $N_d$ increases can lead to an increase in CLWP. The simulation results of Zhao et al. (2024) also indicated that in precipitating clouds, CLWP shows a relatively stable increase with the increase in $N_d$, whereas in non-precipitating clouds, CLWP first increases and then decreases with the increase in $N_d$. To gain a more comprehensive understanding of

the impact of aerosols on liquid-phase clouds, it is necessary to conduct comparative analyses of aerosol-cloud processes under different meteorological and aerosol backgrounds.

When investigating the influence of aerosols on clouds, observations-based studies are limited by deficiencies in the completeness and spatiotemporal continuity of observations (Rosenfeld et al., 2014; Seinfeld et al., 2016) and retrieval biases (Gryspeerdt et al., 2016; Wagner and Kleiss, 2016; Arola et al., 2022), making the combination of observations

with studies using numerical models necessary. This study employs the WRF-Chem-SBM model that couples the online-chemistry version of the Weather Research and Forecasting model (WRF-Chem) and the spectral bin cloud microphysics (SBM) scheme (Gao et al., 2016). This model uses discrete bins to diagnose aerosols and hydrometeors online, allowing for precise treatment of aerosol-cloud microphysical and chemical processes. The assessment based on multi-source observations indicates that such an SBM model performs better than models using bulk cloud parameterizations (Khain

et al., 2015; Zhang et al., 2021), showing strong performance in reproducing meteorological conditions, aerosols, and cloud parameters.

Over land, fast surface temperature variations and topographic influences lead to dynamical and thermal instabilities in the atmospheric boundary layer, which in turn also have important effects on clouds (Choudhury et al., 2019; Liu et al., 2022). In contrast, sea surface temperature varies slowly and is not affected by topography, making the marine

atmosphere relatively stable and more conducive to the formation and development of stable and aerosol-susceptible liquid-phase clouds such as stratocumulus clouds (Wood, 2012). Focusing on marine liquid-phase clouds allows for relatively accurate separation and quantification of aerosol effects on clouds. Eastern China (EC) is one of the major anthropogenic aerosol emission regions over the world. Large-scale subsidence during the winter monsoon leads to capping inversions over the boundary layer of the EC's adjacent ocean (ECO), providing favorable conditions for liquid-

phase clouds (Chang et al., 2021). As shown in the statistics of Niu et al. (2022), low clouds over ECO occur most frequently in winter, exceeding 70%, with stratocumulus as the dominating cloud type. This, combined with the transport of massive continental aerosols from EC to ECO under winter monsoon, makes the analysis of aerosol-cloud interactions for liquid-phase clouds over ECO in winter an ideal scenario for quantifying the impact of aerosols on clouds.

## 2 Methods and data

2.1 Model description and setup

This study utilizes the WRF-Chem-SBM model, which couples the Model for Simulating Aerosol Interactions and Chemistry (MOSAIC) aerosol module (four-bin) of the WRF-Chem model and the SBM scheme, to perform aerosol-cloud simulation, enabling online simulation and bi-directional feedbacks between aerosols and clouds (Gao et al., 2016). In this model, the MOSAIC aerosol module treats the mass and number distributions of the nine main aerosol species

(sulfate, nitrate, chloride, ammonium, sodium, black carbon, primary organics, other inorganics, and water). The aerosol particles are assumed to be internally mixed (Zaveri et al., 2008). The diameters of the four bins range from 0.039-0.156,



0.156-0.624, 0.624-2.5 and 2.5-10.0 μm, respectively. The module treats processes such as aerosol emissions, gas-particle transition, coagulation, in-cloud liquid-phase chemistry, dry deposition, and wet scavenging (Sha et al., 2019; Sha et al., 2022). The SBM scheme in this model is the fast version, which solves a system of prognostic equations for three

hydrometeor types (droplets, ice/snow, and graupel) and CCN, by numerically discretizing their size distributions. Each size distribution function is structured into 33 mass-doubling bins, wherein the mass within the k-th size bin is twice that within the (k-1)-th bin. This scheme can treat cloud microphysics processes such as aerosol activation, freezing, melting, diffusion growth/evaporation of droplets, deposition/sublimation of ice particles, droplets and ice collisions (Khain et al., 2004).

In the simulations, we adopt nested grids with resolutions of 12 km and 2.4 km, as shown in Fig. 1. The outer domain is centered at (32°N, 120°E) with a grid number of 151×125, while the inner domain covers the ECO region with a grid number of 121×121 (highlighted by the red box in Fig. 1). The simulation period is from 00:00 on Feb. 1, 2019, to 00:00 on Feb. 5, 2019, in UTC. The initial 12 hours are considered as model spin-up and are not included in the analysis. Meteorological initial and boundary conditions are from the National Center for Environmental Prediction (NCEP) FNL

reanalysis with a temporal and spatial resolution of 6 h and 0.25°, respectively (NCEP et al., 2015). Chemical initial and boundary conditions are from the Community Atmosphere Model with Chemistry (CAM-chem, Buchholz et al., 2019; Emmons et al., 2020). Anthropogenic emissions are from the Multi-resolution Emission Inventory for China (MEIC, 2016 version developed by Tsinghua University (Li et al., 2017; Zheng et al., 2018). The model parameterization settings are listed in Table 1. In addition, a four-dimensional assimilation method is used in this study to improve the model's

ability to simulate the meteorological field and thus enhance the model's ability to reproduce the factual aerosol-cloud scenario (Zhao et al., 2020; Hu et al., 2022). The observations used in the assimilation are from the NCEP operational global observation subsets at surface (NCEP et al., 2004) and upper air (Satellite Services Division et al., 2004).

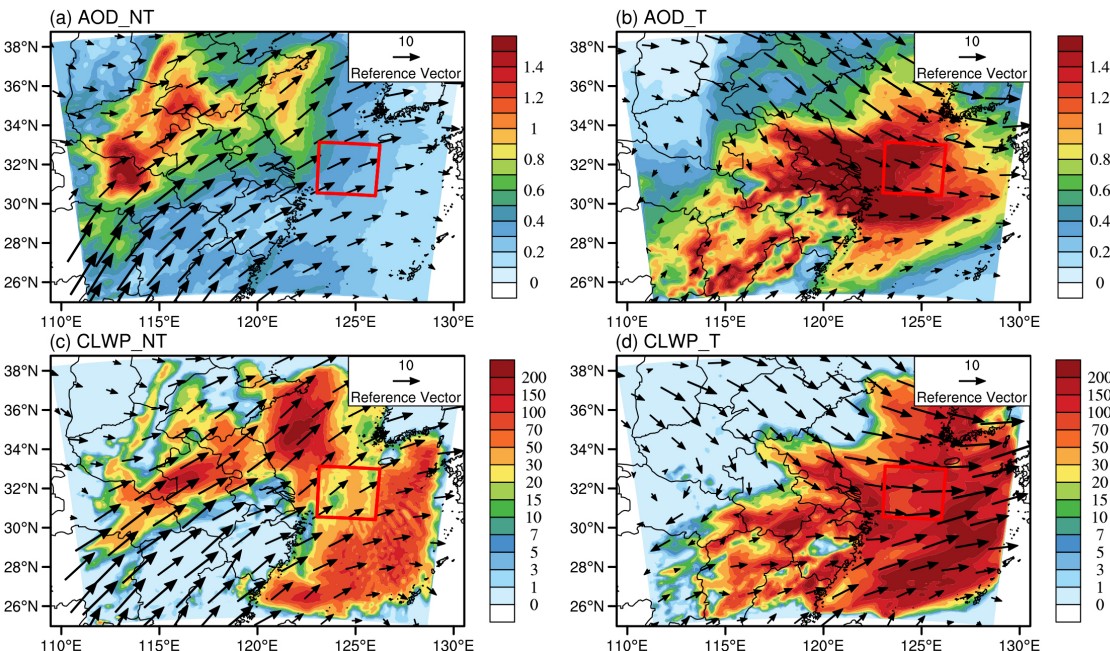

**Figure 1.** Simulation domain and distribution of AOD, CLWP (in g·m⁻²), and 850 hPa wind field (in m·s⁻¹) during NT (a and c) and T (b and d). AOD and CLWP are from the Control experiment, a-b represent MICAPS wind fields, c-d



represent simulated wind fields from the Control experiment.

**Table 1.** Model parameterization settings. "Number" refers to the WRF-Chem namelist switch.

| Process | Number | Name |
|---|---|---|
| Longwave radiation | 4 | RRTMG (Mlawer et al., 1997) |
| Shortwave radiation | 4 | RRTMG (Iacono et al., 2008) |
| Surface layer | 1 | MM5 Monin-Obukhov (Pahlow et al., 2001) |
| Land surface | 2 | Unified Noah (Chen et al., 2010) |
| Boundary layer | 1 | YSU (Shin et al., 2012) |
| Chemistry and aerosols | 9 | CBMZ and four-bin MOSAIC (Sha et al., 2022) |
| Photolysis | 2 | Fast-J (Wild et al., 2000) |
| Sea salt emission | 2 | MOSAIC/SORGAM (Fuentes et al., 2011) |
| Dust emission | 13 | GOCART (Zhao et al., 2010) |
| Biogenic emission | 3 | MEGAN (Guenther et al., 2006). |

In order to qualitatively and quantitatively analyze the effect of aerosols on clouds, two experiments are set up in this study, namely the Control experiment for the factual scenario and the Sen experiment (counterfactual scenario) in which the continental aerosol (aerosols other than sea salt) emissions are turned off. Additionally, we select two periods

of EC aerosol non-transiting (NT, 12:00 Feb. 1 - 12:00 Feb. 2, 2019) and transiting (T, 0:00 Feb. 3 - 0:00 Feb. 4, 2019) ECO within the simulation period to analyze the cloud response to aerosols under different aerosol and meteorological conditions through a comparative analysis. According to the simulations (Fig. 1 a-b), during the NT period, ECO is located on the east side of the EC continental aerosol transiting path, and the aerosols mainly come from the diffusion of EC aerosols and the long-distance transportation of aerosols from southern China. In consequence, the aerosol

concentration is relatively low, with a regional average aerosol optical depth (AOD) of 0.33. During T, ECO is located on the main transport path of EC aerosols and is close to the EC aerosol emission hot-spot, resulting in high aerosol concentrations in ECO, with a regional mean AOD as high as 1.43.

2.2 Observational data

We utilize various observational datasets for meteorological fields, aerosols, and clouds, to evaluate the simulation results. Precipitation data is obtained from the Integrated Multi-satellitE Retrievals for GPM (IMERG) dataset (Huffman et al., 2019), of which the daily accumulated high quality precipitation product is used in this study, with a temporal and spatial resolution of 1 day and 0.1°, respectively. Other meteorological data, including temperature, dew point depression, and wind field, are obtained from the Meteorological Information Comprehensive Analysis and Processing System

(MICAPS, Hu et al., 2018) developed by the National Meteorological Center of China (NMC). The data have temporal and spatial resolutions of 12 hours and 2.5°, respectively, and include 11 vertical layers.

For aerosol, we use near-surface observations of $PM_{2.5}$ and satellite-retrieved AOD data. The near-surface $PM_{2.5}$ data is obtained from the National Urban Air Quality Real-time Release Platform of China National Environmental Monitoring Center, which contains hourly surface observations from more than 1600 stations. The AOD data is obtained

from the MOD04_L2 dataset of the Moderate Resolution Imaging Spectrometer (MODIS, Levy and Hsu, 2015), of which the AOD product combining the "Dark Target" and "Deep Blue" algorithms with temporal and spatial resolutions



of 5 min and 10 km, respectively, is used in this study.

The cloud parameters, containing CER, $N_d$, cloud optical thickness (COT), and CLWP, are taken from the MOD06_L2 daily dataset (Platnick et al., 2015) with a spatial resolution of 1 km. In addition, since $N_d$ is not included in

the MODIS dataset, we use MODIS COT and CER to calculate the column-integral $N_d$ based on the approach of Han et al. (1998, see also Brenguier et al. (2000)):

$$N_d = \gamma \cdot COT^{0.5} \cdot CER^{-2.5} \tag{1}$$

where $\gamma$ is a constant valued at $1.37 \times 10^{-5}$ m$^{-5}$ (Quaas et al., 2006).

2.3 Data processing

Due to the resolution differences between the observations and simulations, interpolation is needed before comparing them. Specifically, for observational data with higher resolution than the model grid, the observational data is interpolated to the model grid, while for observational data with lower resolution than the model grid, the simulation data is interpolated to the grid of the observational data.

When comparing the spatial distribution of simulation results with satellite-retrieved data averaged over the study period, it is unreasonable to directly compute the average of the simulation results for the entire period due to the spatio-temporal discontinuities of satellite-retrieved data. Therefore, to make the comparison with the satellite-retrieved data, spatio-temporal matching is performed for the simulation data. Specifically, only when satellite data at a given grid point and time is valid, the corresponding simulation value is included in the average calculation.

In addition to the above, further processing is required for both satellite-retrieved and simulated liquid-phase cloud parameters. For MODIS cloud parameters, cloud retrievals with high reliability are selected using the method referenced from Saponaro et al. (2017): (1) Liquid-phase cloud data are chosen based on MODIS cloud phase parameters, and (2) transparent-cloudy pixels (COT < 5) are screened out to limit uncertainty. When calculating the simulated cloud parameters for a specific grid point, in cases where multiple independent clouds exist at different altitudes, we follow the

method of the instrument simulator, treating these clouds as if they are from a single homogeneous layer (Pincus et al., 2012). When comparing the simulations with satellite-retrieved cloud parameters, we refer to the processing of satellite-retrieved cloud parameters and the threshold method used by Roh et al. (2020) for distinguishing cloud phases. The following criteria are applied to filter the simulated cloud parameters: (1) cloud optical thickness of water (COTW) > 0.1 and cloud optical thickness of ice (COTI) < 0.01 for each layer, and (2) column COTW ≥ 5. The simulated COTW and

COTI used in this study are calculated based on the methodology in the Goddard radiation scheme (Chou and Suarez, 1999) of the WRF-Chem model. Additionally, in the evaluation, the method used to calculate simulated $N_d$ is the same as that used for satellite-retrieved $N_d$. In the analysis of the simulation results, $N_d$ is directly taken from the model output, and the criteria for liquid-phase clouds are strictly defined as cloud liquid water content (CLWC) > 0 and cloud ice water content (CIWC) = 0.

When analyzing the impact of aerosols on liquid-phase clouds in ECO based on model results, we use two different averaging methods when calculating vertical weighted averages, temporal averages, or spatial averages of aerosol and cloud parameters. In Sect. 3.2, we focus on the overall impact of continental aerosols on liquid-phase clouds in ECO, and directly calculate the average, where the values from non-liquid-phase cloud vertical layers and grid points, set to zero, are also included in the calculation. In Sect. 3.3, we focus on the response of cloud parameters to aerosols under different

aerosol and meteorological conditions. To ensure the accurate capture of this response signal, we only calculate the average within the liquid-phase clouds, meaning the values from non-liquid-phase cloud vertical layers and grid points are excluded from the calculation of the average (cloud-sky values).

The analysis in this study is based on high-resolution simulations in the inner domain, but the outer domain results



are used when evaluating meteorological and aerosol simulations due to the limitations in resolution (the MICAPS meteorological field has relatively coarse resolution) and availability ($PM_{2.5}$ data is only available from land-based observations) of the observational data. In addition, as the IMERG data we use is daily cumulative precipitation, the IMERG and simulated daily cumulative precipitation for February 2 is used to evaluate precipitation during the NT period, rather than the previously defined 12:00 February 1 to 12:00 February 2. When evaluating cloud simulations, the inner domain results are directly assessed due to the relatively high resolution of the MODIS data.

# 3 Results and discussion

3.1 Evaluation of simulation results

To ensure the reliability of the research, various observational data are used to evaluate the simulated meteorological, aerosol, and cloud parameters. A comparison of temperature, dewpoint depression, and wind profiles between model and MICAPS data shows good agreement (Fig. S1), as expected thanks to the data assimilation. In consequence, the model (Control experiment) can effectively treat the processes of aerosol generation, transport, and deposition, simulating AOD and near-surface $PM_{2.5}$ with high agreement both numerically and in distribution with observations (Fig. 2). Although there are only sparse MODIS AOD retrievals over the region of interest, it is evident from the adjacent grid boxes that the model represents the elevated AOD during T, compared to NT, with a similar factor as the MODIS retrievals do. $PM_{2.5}$ measurements are available only over land, but the larger $PM_{2.5}$ over land during T compared to NT is similar between simulation and observations. In the Sen experiment results excluding continental aerosols, the only small amount of ocean emissions leads to low AOD.

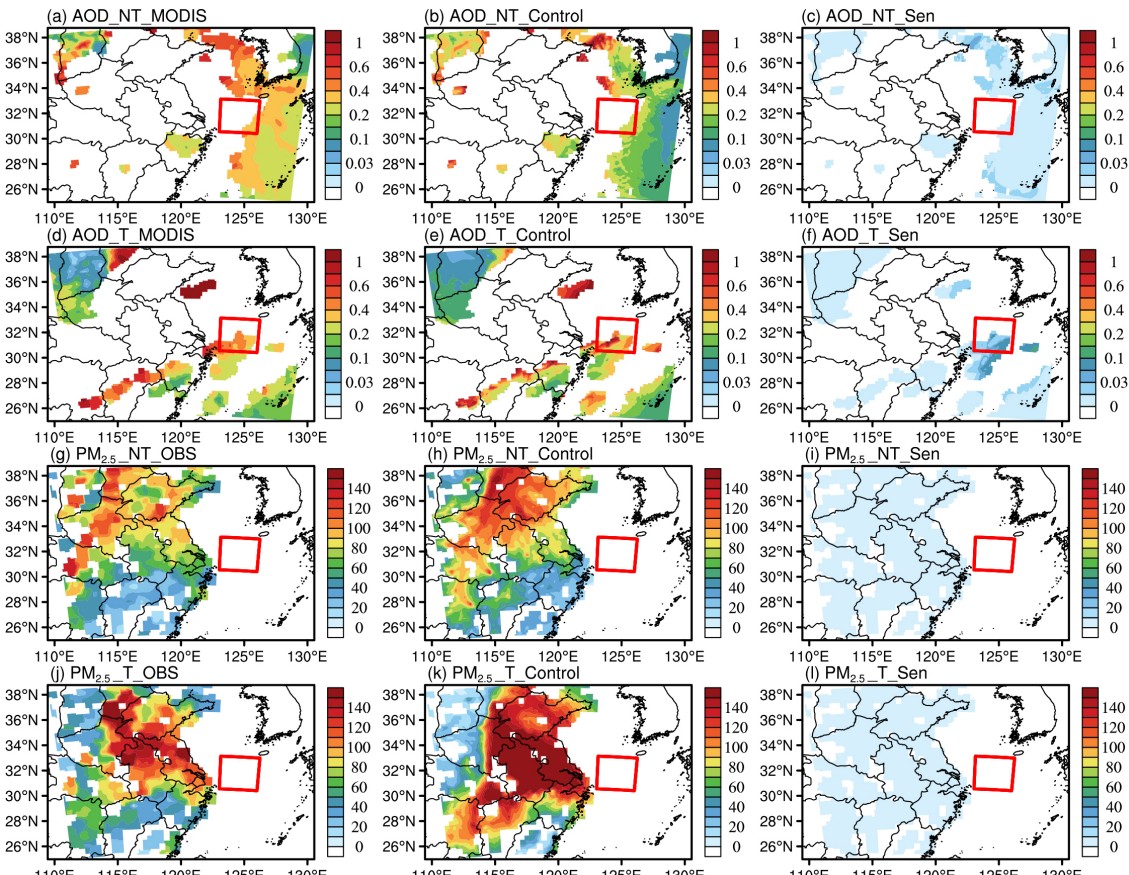

**Figure 2.** Observed (left column, AOD and near-surface PM$_{2.5}$ from MODIS retrievals and near-surface observations, respectively) and simulated (middle and right columns, from Control and Sen experiments, respectively) AOD (a-f) and near-surface PM$_{2.5}$ (g-l, in μg·m$^{-3}$) during NT (a-c and g-i) and T (d-f and j-l). Model output is shown only for columns where observations are available.

Some evaluation of cloud parameters and precipitation simulation is presented in Fig. 3. As shown in the results of the Sen experiment, in the low aerosol concentration environment with only oceanic emissions, the clouds are mostly thin with low N$_d$ and CLWP (some irregularly shaped blanks in the figure are mainly caused by filtering out grids with low COT), and there is widespread weak precipitation. Compared to the Sen experiment, the Control experiment simulating the factual scenario shows a significant increase in both N$_d$ and CLWP, and the weak precipitation in the Sen experiment is largely suppressed, resulting in more concentrated precipitation hot-spots. Although the results of the Control experiment exhibit some differences from satellite retrievals, such as failing to reproduce the clouds in the southwestern part of the domain during T and a slight shift in the simulated precipitation location, the model generally reproduces the values and distribution of the satellite retrievals.





**Figure 3.** Distributions of satellite-retrieved (left column) and simulated (middle and right columns, from the Control and Sen experiments, respectively) $N_d$ (a-f, in $cm^{-3}$), CLWP (g-l, in $g \cdot m^{-2}$), and cumulative precipitation (m-r, in mm) during NT (a-c, g-i, and m-o) and T (d-f, j-l, and p-r). Satellite retrievals are from the MODIS instrument (Platnick et al., 2017)





for the cloud properties, and from IMERG for precipitation. The model is sampled only for columns where satellite retrievals are available.

Overall, the model reasonably reproduces the observed meteorological, aerosol, and cloud parameters for both periods, providing a basis for confidence in the model-based aerosol-cloud analysis.

3.2 The impact of aerosols on cloud microphysics in different environments

The hypothesis of the following analysis is that aerosols exert different effects on clouds in different environments. A comparative analysis during the two periods implying different meteorological backgrounds and two experiments

presenting different aerosol burdens was conducted to discuss the impact of aerosols on cloud microphysics in various environments. As shown in Fig. 4, during T, a relatively low vertical temperature lapse rate and a relatively high water vapor content are shown in ECO. In contrast, during NT, the vertical temperature lapse rate from the surface to 1000 m is higher than that during T, primarily due to the lower temperatures around 1000 m altitude. The more unstable atmosphere leads to stronger vertical motion during NT, particularly at altitudes between 700 and 1000 m, where vertical wind speeds

are significantly higher than during T. Based on relatively lower temperatures and stronger vertical motion, the relative humidity (RH) at altitudes of 900-1100 m during NT is significantly higher than during T, even though the water vapor content is lower. At other altitudes, RH during NT is lower than during T. In addition, the sea surface wind speed during T is 1.5 times that during NT, resulting in more sea salt emissions during T. However, whether during NT or T, the amount of aerosols emitted from the ocean is negligible compared to the continental aerosols transported over, with the

former being three orders of magnitude lower than the latter.

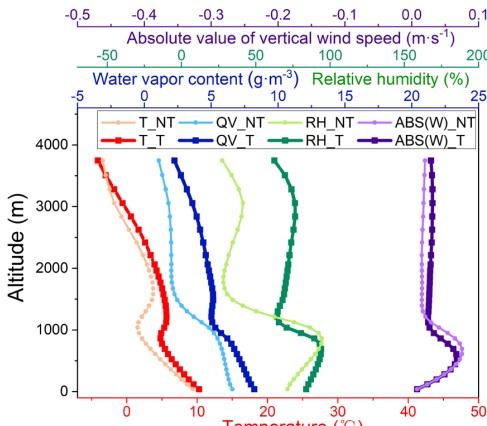

**Figure 4.** Vertical profiles of temperature (T), water vapor content (QV), relative humidity (RH) and absolute value of vertical wind speed (ABS(W)) from the Control experiment during NT and T.


As shown in Fig. 5, in the absence of continental aerosols (Sen experiment), sea salt aerosols mainly appear from near surface to 1000 m altitude. During both periods, cloud droplets and cloud liquid water are found primarily around 500 m. However, the relatively unstable atmosphere during NT results in higher cloud top heights compared to during T, while the higher water vapor and aerosol contents during T lead to higher $N_d$ and CLWP than during NT. In the case of

only sea salt aerosols, aerosols in clouds can be fully activated (Fig. 6a), and the activated aerosols in ECO show a spectral distribution largely parallel to that of the total aerosols. More ocean emissions during T lead to more sea salt



aerosols entering the clouds, which are activated to form more cloud droplets than during NT, however, this also leads to an increase in small cloud droplets and a decrease in large droplets (Fig. 6b). In the factual scenario (Control experiment), most of the aerosols in ECO originate from continental emissions. During T, aerosols in ECO exhibit a vertical

concentration gradient that decreases from low to high altitudes (Fig. 5), similar to that near the source area. However, during NT, aerosols in ECO mainly come from diffusion, and aerosols exhibit a nearly uniform distribution from near surface to 1300 m altitude, showing a slight variation of a high-low-high vertical distribution. During both periods, continental aerosols led to an increase in $N_d$, a decrease in CER, and an increase in CLWP and cloud top height in ECO (Figs. 5 and 6b). The difference lies in that during NT, continental aerosols lead to higher cloud top heights compared to

during T, but they also cause a significant reduction in CLWC at altitudes below 700 meters. During T, continental aerosols result in increased CLWC at all layers, with a much greater increase in CLWP than during NT. In addition, as shown by the aerosol and cloud droplet spectrum distribution (Fig. 6), compared to the Sen experiment, where the total aerosol and activated aerosol exhibit nearly parallel spectral distributions, the Control experiment shows a significant decrease in the activation proportion of the first bin aerosols. This is because the environment could not meet the high

supersaturation requirements for fully activating such a large number of small aerosols. Moreover, compared to the Sen experiment, where the increase in $N_d$ during T relative to NT results in more small droplets but fewer large droplets, both large and small cloud droplets increase significantly during T compared to NT in the Control experiment. Overall, under relatively unstable and dry atmospheric conditions, aerosols can raise cloud top heights but only have a weak effect on cloud thickness and CLWP. In contrast, under relatively stable and humid atmospheric conditions, aerosols significantly

enhance all three cloud parameters.

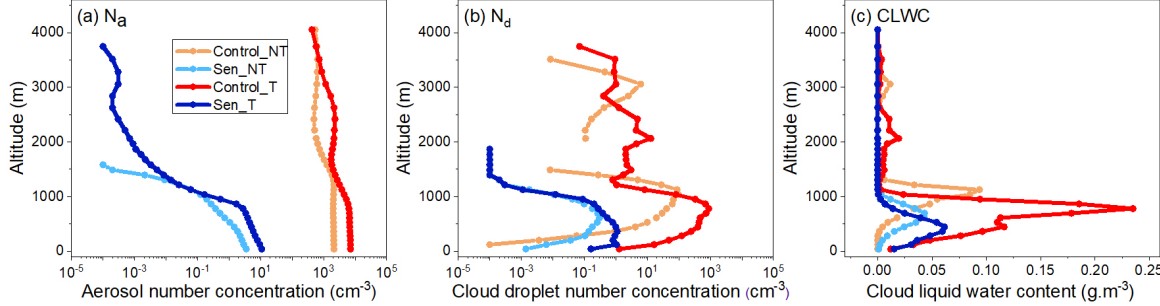

**Figure 5.** Vertical profiles of $N_a$ (a), $N_d$ (b) and CLWC (c) from the Control and Sen experiments during NT and T.

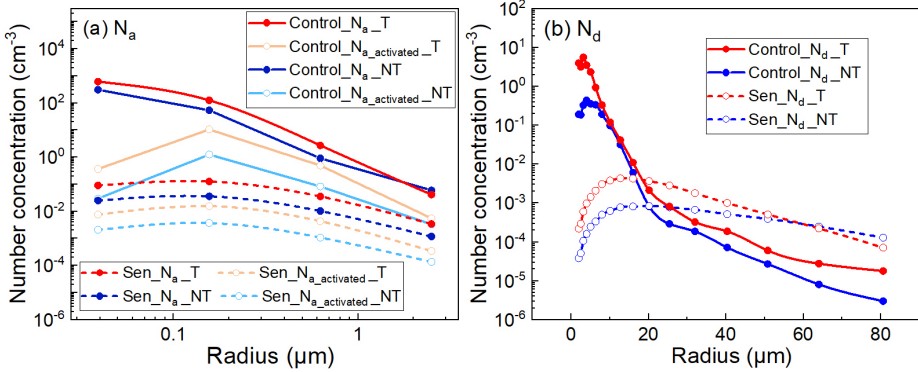


**Figure 6.** Aerosol (a, $N_a$ is the total aerosol number concentration, $N_{a\_activated}$ is the activated aerosol number





concentration) and cloud droplet (b) spectral distributions of the Control and Sen experiments during NT and T.

Continental aerosols have a significant impact on precipitation in ECO (Fig. 7). In the absence of continental
aerosols, during the more unstable NT with stronger vertical motion, the rainwater path (RWP) in ECO (4.7 g·m⁻²) is
much higher than during T (2.2 g·m⁻²). However, in the environment with high aerosol concentrations that includes
continental aerosols, $N_d$ increases, and cloud droplet sizes decrease to below the precipitation threshold, leading to
precipitation being significantly suppressed in areas with high rainwater content (RWC) in the Sen experiment. As a
result, liquid-phase clouds need to develop much deeper for raindrops to form through condensation and collision. The
relatively unstable and dry environment during NT results in relatively thin cloud layers, making it difficult for
precipitation to form. In contrast, the relatively stable and humid environment during T allows the clouds to develop
deeply, leading to relatively strong precipitation.

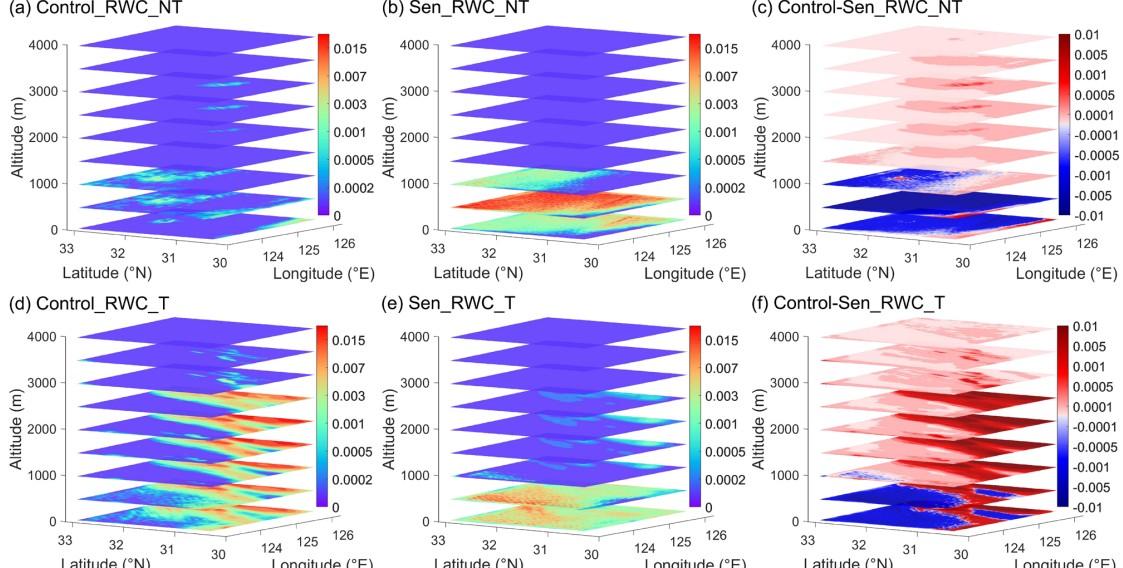

**Figure 7.** Distributions of average RWC (in g·m⁻³) in liquid-phase clouds at each altitude during NT (a-c) and T (d-f)
from the Control (left column) and Sen (middle column) experiments, and the difference between the two experiments
(right column).

3.3 Variations in the role of factors influencing aerosol-cloud interactions in different environments

In different environments, the effects of aerosols and meteorological fields on aerosol-cloud interactions may also
vary. We selected four representative areas, each characterized by varying intensities of aerosols, clouds, and
precipitation, covering an area of 0.2° × 0.2° (the locations and conditions of these areas are shown in Fig. S2). By
comparing the temporal variations of various parameters in different areas, we further discuss the impacts of aerosol and
meteorological factors. We select RH and lower tropospheric stability (LTS) to represent meteorological conditions.
Since clouds during the study period mainly develop between the near-surface and 1300 m altitude, RH is computed as
the vertical weighted average from the near surface to 1300 m, while LTS is calculated as the difference between the
potential temperature at 700 hPa and at the surface (Klein and Hartmann, 1993).



As shown in Fig. 8, RH exhibits a relatively clear negative correlation with LTS over the four areas during NT and the following 12 hours. In contrast, LTS is relatively high, and there is no significant correlation between RH and LTS during T, suggesting that changes in RH during this period are mainly driven by horizontal variations in temperature and water vapor content. In the absence of continental aerosols, during the cloud development and mature stages (mainly reflected in CLWP), $N_d$ and $N_a$ exhibit a positive correlation, while the negative correlation between $N_d$ and CER is weak, with synchronous variations between the two observable during some periods. Compared to clean conditions, in high-aerosol-concentration environments containing continental aerosols, while $N_d$ and $N_a$ generally remain positively correlated, this relationship shows greater variability due to meteorological influences. The changes in supersaturation caused by atmospheric vertical and horizontal motions are the primary factor directly influencing the correlation between $N_d$ and $N_a$. When RH (due to the transient and localized nature of supersaturation, RH is used to represent the overall supersaturation intensity in this environment) is relatively high, $N_d$ and $N_a$ exhibit a clear positive correlation. However, when RH is low, the environment cannot adequately activate aerosols within clouds, limiting the increase in $N_d$.

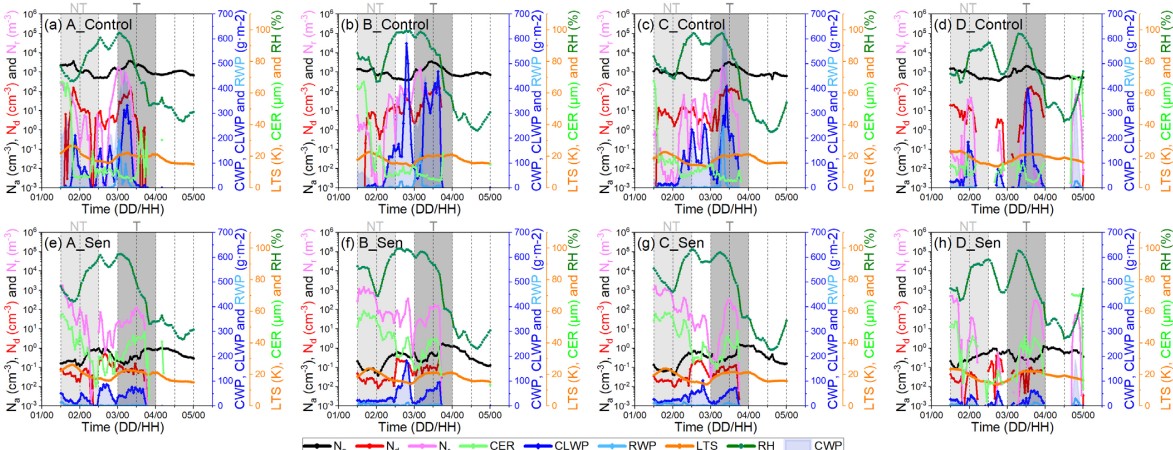

**Figure 8.** Temporal variations of meteorological parameters (RH and LTS), in-cloud aerosol ($N_a$) and cloud parameters ($N_d$, $N_r$, CWP, CLWP, RWP, and CER) in areas A (a and e), B (b and f), C (c and g), and D (d and h) from the Control (a-d) and Sen (e-h) experiments (CWP represents the total cloud water content, including liquid-phase, mixed-phase, and ice-phase cloud water. The light gray and dark gray shaded areas represent NT and T, respectively).

By serving as CCN and influencing cloud droplets, continental aerosols further affect the development of liquid-phase clouds. Overall, while continental aerosols lead to a decrease in CLWP in some conditions, they generally have a positive impact on CLWP. For thin clouds in a relatively dry atmosphere, such as during the first 12 hours of NT in area C, continental aerosols lead to an increase in $N_d$ but a significant decrease in CER. This results in a larger surface area and volume ratio of cloud droplets, enhancing evaporation and reducing CLWP. The additional continental aerosols also accelerate a transition to ice-phase and mixed-phase clouds by elevating cloud top heights, for example, during the first 12 hours in areas A and B. The positive effect of continental aerosols on liquid-phase clouds is most pronounced in high RH environments conducive to cloud development, but varies under different LTS conditions: 1) for low LTS conditions, such as in areas A, B, and C from 12:00 on the 2nd to 00:00 on the 3rd, continental aerosols cause a relatively brief and explosive increase in CLWP. 2) For high LTS conditions, such as during T in the four areas, continental aerosols lead to relatively prolonged high values of CLWP (around 10 hours) in ECO. These changes occur only during cloudy periods in the Sen experiment, indicating that during the study period, continental aerosols primarily alter the intensity of cloud



development without significantly affecting the cloud lifetime, which is dominated by the meteorological conditions.

In terms of precipitation, during the relatively dry NT, continental aerosols cause reductions in both the number concentration of raindrops ($N_r$) and RWP. Under high RH conditions, the major precipitation enhancement caused by continental aerosols occurs during T with high LTS. Due to weak convection in winter, under low LTS conditions, such as from 12:00 on the 2nd to 00:00 on the 3rd in area B, even though there is an explosive increase in CLWP, the growth of RWP is very limited. Furthermore, since precipitation from liquid-phase clouds in winter over ECO is relatively weak,

even in the precipitation hot-spot area C, the daily cumulative precipitation is less than 10 mm, resulting in minimal effects from wet removal during the study period. A comparison of $N_a$, $N_d$, and RWP in areas C and D during T shows that the strongest precipitation in ECO during the study period causes only a halt in the increase of $N_d$ from 06:00 to 08:00 on the 3rd in area C, without significantly affecting the overall changes in $N_d$, and its influence on $N_a$ is comparatively weaker than on $N_d$.

These changes occur only during the cloudy periods in the Sen experiment, indicating that during the study period, continental aerosols primarily alter the intensity of cloud development without significantly affecting the cloud lifetime dominated by the meteorological fields.

## 4 Summary and conclusion

    Based on simulations with the WRF-Chem-SBM model, in which spectral bin cloud microphysics and an online

aerosol module are coupled, the effects of aerosols on liquid-phase clouds in ECO under different meteorological and aerosol backgrounds are analyzed. We conducted a comprehensive model evaluation using various data for meteorology, aerosol, and cloud. The evaluation shows that, with the four-dimensional data assimilation, the model accurately reproduces the observed meteorological fields. With this support, the simulated aerosol parameters align well with observations in both magnitude and distribution. Due to the complexity of simulating clouds and precipitation, there are,

however, noticeable differences between the simulations and observations. For instance, some of the clouds shown in MODIS data are not well simulated, and the simulated rainbands exhibit a slight displacement compared to IMERG data. However, the model overall reproduces the observational cloud fields, providing a basis for the credibility of analyses based on the simulations.

    By comparing the differences in aerosol and cloud parameters between the Control (factual scenario) and Sen

(counterfactual scenario in which continental aerosol emissions are disabled) experiments and between T (relatively stable and moist) and NT (relatively unstable and dry) periods, we analyze the impact of aerosols on cloud properties in different environments. In the absence of continental aerosols, the different meteorological conditions result in relatively low cloud top heights during T and relatively high cloud top heights during NT. However, whether during T or NT, $N_a$ is a dominant factor in $N_d$ variations. When including continental aerosols, the atmosphere fails to enable full activation of

aerosols during both periods, and the effects of aerosols on $N_d$ are significantly modulated by meteorological factors. During NT, continental aerosols elevate cloud top heights but reduce CLWC at low altitude, without a noticeable increase in cloud thickness. In contrast, during T, continental aerosols show a significant increase in both cloud top height and thickness. Additionally, under clean conditions, convection is stronger during NT, resulting in more precipitation compared to T. However, continental aerosols suppress precipitation shown in clean conditions in both periods, with a

strong precipitation only occurring during T through the development of deep clouds. This suggests that a stable but humid atmosphere is more conducive to cloud development and the positive influence of aerosols on clouds and precipitation compared to an unstable but dry atmosphere.

    To further discuss the impact of various factors on aerosol-cloud interactions in different environments, we examine the temporal variations of meteorological, aerosol, and cloud parameters in the four representative areas. For the winter

liquid-phase clouds in ECO during the study period, the meteorological fields determine the cloud lifetime, while

aerosols only affect clouds during their development stage without noticeably impacting their lifetime. Compared to the clean conditions where $N_a$ primarily drives $N_d$ variation, the influence of continental aerosols on $N_d$ is largely modulated by supersaturation, with low supersaturation limiting the full activation of continental aerosols. For thin cloud layers in relatively dry atmospheres, the increase in $N_d$ caused by continental aerosols reduces cloud droplet size, which in turn

enhances evaporation and reduces CLWP. By elevating cloud top height, continental aerosols facilitate the transition from liquid-phase clouds to mixed-phase or ice-phase clouds. The enhancing effect is more significant in high RH environments and varies under different LTS: 1) under low LTS, continental aerosols lead to a relatively brief and explosive increase in CLWP. 2) Under high LTS conditions, continental aerosols cause relatively prolonged periods of high CLWP. Since precipitation in this study is relatively weak, with daily cumulative rainfall in the precipitation hot-

spots below 10 mm, wet scavenging during the most intense precipitation periods only halts the increase in $N_d$ and does not have a significant effect on its overall variation.

*Code and Data availability.* The WRF-Chem model code can be downloaded from https://www2.mmm.ucar.edu /wrf/users/download/get_sources.html. The WRF-Chem–SBM model code can be obtained by contacting Jiwen Fan of

Argonne National Laboratory. The model outputs are available upon request. The MICAPS wind field data comes from the NMC (http://www.nmc.cn, last access: 29 February 2024). IMERG precipitation data can be obtained from https://disc.gsfc.nasa.gov/datasets/GPM_3IMERGDF_06/summary?keywords=Precipitation (last access: 29 February 2024). The near-surface $PM_{2.5}$ data comes from the National Urban Air Quality Real-time Release Platform of the China National Environmental Monitoring Centre (https://air.cnemc.cn:18007, last access: 29 February 2024). MODIS AOD

and cloud parameters can be obtained from https://lad sweb.modaps.eosdis.nasa.gov/search/order/1/MOD06_L2-61 and https://ladsweb.modaps.eosdis.nasa.gov/search/order/1/MOD06_L2--61 respectively (last access: 29 February 2024).

*Author contributions.* JZ and XM designed and conducted the model experiments, analyzed the results, and wrote the paper. XM developed the project idea and supervised the project. XM, and JQ proposed scientific suggestions and revised

the paper.

*Competing interests.* One of the (co-)authors is a member of the editorial board of Atmospheric Chemistry and Physics, and the authors have no other competing interests to declare.

*Acknowledgements.* The numerical calculationsin this paper were conducted in the High-Performance Computing Center of Nanjing University of Information Science & Technology. We express our gratitude to Dr. Jiwen Fan of Argonne National Laboratory for providing the code for the WRF-Chem–SBM model. We are grateful to the National Aeronautics and Space Administration, the National Centers for Environmental Prediction, the MEIC support team, the Chinese National Meteorological Center, and the China National Environmental Monitoring Center for providing the MODIS and

GPM data, FNL and observation subsets, MEIC emission inventory, MICAPS data, and $PM_{2.5}$ data, respectively.

*Financial support.* This research has been supported by the Second Tibetan Plateau Scientific Expedition and Research program (grant no. 2019QZKK0103), the National Natural Science Foundation of China (grant nos. 42061134009 and 41975002), the China Scholarship Council program (grant no. 202309040034), and the Postgraduate Research and

Practice Innovation Program of Jiangsu Province (grant no. KYCX22_1151).

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
