# Peer review of "Aerosol-cloud interactions in liquid-phase clouds under different"

_EGUsphere, 2024_

## Author Comment (AC1)

**Response to the Comments of Referees**

**Journal:** Atmospheric Chemistry and Physics
**Manuscript Number:** egusphere-2024-3662
**Title:** Aerosol-cloud interactions in liquid-phase clouds under different meteorological and aerosol backgrounds
**Author(s):** Jianqi Zhao, Xiaoyan Ma, and Johannes Quaas

We thank the reviewers and editor for providing helpful comments to improve the manuscript. We have revised the manuscript according to the comments and suggestions of the referees.

The referee's comments are reproduced (black) along with our replies (blue). All the authors have read the revised manuscript and agreed with the submission in its revised form.

**Anonymous Referee #1**

This manuscript shows real-world simulations of a cloud-aerosol interaction case and compares it to observations. The simulations are sophisticated, e.g., they use bin microphysics. The simulations are potentially informative, but I had difficulty understanding the manuscript. E.g., several times, the authors claim that one of their figures shows some phenomenon, but when I look at the figure, I don't see that phenomenon. Some instances are listed below. Maybe the figures need to be redesigned, for instance, split up and simplified. In addition, while there are lots of words describing what is (purportedly) plotted in the figures, a key issue of how the aerosol affects rainfall is not convincingly explained.

Thank you very much for your comments. We have made substantial revisions to the manuscript, including re-running the model with more appropriate configurations, revising the figures to improve readability, providing correspondence and statistical analysis for the text, and providing more reasonable explanations for some of the content. Details of the revisions can be found in the following responses.

Abstract: Please state what type of cloud you'll be exploring. Stratocumulus? Cumulus? Shallow clouds? Deep clouds?

The reviewer raises a good question, and we clarified the scope now early on. We study liquid-phase clouds of different kinds, and analyze the effect of aerosols on the liquid-phase cloud in different environments. We thus do not make a more detailed classification of liquid-phase clouds.

Abstract: The abstract leads the reader to believe that the manuscript will discuss liquid-only clouds: "We conduct a comparative analysis of aerosol-cloud responses in liquid-phase clouds". But only a few sentences later it talks about mixed-phase clouds "the transition of liquid-phase clouds into mixed-phase or ice-phase clouds." Can the wording be clarified?

The reviewer has a good point. Previously we were to discuss some anomalies in the aerosol-cloud relationship due to the transformation of liquid-phase clouds into other phases under the influence of aerosols. In the revised manuscript, we have excluded these cases and analyzed only the liquid-phase

cloud. The manuscript is revised to clarify this.

Line 48: "in non-precipitating clouds, CLWP first increases and then decreases with the increase in Nd." This is somewhat similar to the result found by Chen et al. (2024)'s paper entitled "Magnitude and timescale of liquid water path adjustments to cloud droplet number concentration perturbations for nocturnal non-precipitating marine stratocumulus".

This reference helped us improve the introduction, and we have now cited it.

Line 90: "resolutions of 12 km and 2.4 km". A 2.4-km grid spacing won't be able to fully resolve turbulent updrafts (see Fig. 4), and hence won't activate aerosol accurately. Are subgrid updrafts parameterized in the model? If so, how? Can you make some comments on the accuracy of the simulated vertical velocity?

This was indeed an omission in the previous manuscript and is clarified in the revision. Yes, we used the Grell-3 scheme to treat subgrid convection, and we added this note to Table 1 and provided the namelist file of WRF-Chem-SBM as Table S3 in the supplement.

Due to the difficult nature of the observations, we were unable to acquire usable vertical velocity data. The heights at pressure levels are generally influenced by the static equilibrium and thermal structure of the atmosphere, and this affects the distribution of vertical velocities. To address this, we added the simulated heights at each pressure level to Fig. S1 as an indirect evaluation of the vertical velocities. In addition, the accuracy of the vertical velocity simulation is a key factor in the accuracy of the simulation of atmospheric saturation and aerosol activation, and the evaluation of dewpoint depression and $N_d$ can also, to some extent, corroborate the model's simulation of vertical velocities.

Line 113: "Sen experiment (counterfactual scenario)" I've never before heard the expression "Sen experiment". Is "Sen" an abbreviation for "sensitivity"? If so, please say so.

The reviewer is right that this is not a universally-known abbreviation. Since there is only one sensitivity experiment in this study, the abbreviation "Sensitivity" is used directly to represent the experiment for ease of mention. We added the note 'Sen (abbreviation for "Sensitivity") experiment' at this place.

Fig 5: How does the precipitation profile differ between the NT and T runs? Could you include the precipitation profile in Fig. 5? It looks like the T time period has a greater reduction in CLWP. Is this what one would expect given the theory described in Ackerman et al. (2004)'s paper entitled "The impact of humidity above stratiform clouds on indirect aerosol climate forcing"? (However, in the T case, the RWP decreases when the continental aerosols are omitted, which may differ from what Ackerman found.)

This is a useful suggestion, and we have incorporated the precipitation profile as Fig. 6d in the revised manuscript.

Continental aerosols lead to a greater increase in CLWP during T than NT. Our results do not contradict the findings of Ackerman et al. (2004). Ackerman et al (2004). pointed out that "only

when the overlying air is humid or droplet concentrations are very low does sufficient precipitation reach the surface to allow cloud water to increase with droplet concentrations. Otherwise … cloud water is reduced as droplet concentrations increase." In relatively dry conditions with a higher number of cloud droplets, such as below 800 m during the NT period, an increase in cloud droplets leads to a decrease in CLWC and RWC (as shown in Figs. 6 and 7). However, during the relatively humid T period, or at altitudes where high RH (800–1000 m) and extremely low cloud droplet concentrations (above 1000 m) during the NT period, an increase in cloud droplets results in an increase in CLWC and RWC. These findings are consistent with those of Ackerman et al. (2004). We have revised the last paragraph of Section 3.2 to explain that an excessive number of small cloud droplets suppresses precipitation while also showing that in some areas lacking CCN, continental aerosols can enhance precipitation by providing CCN. Additionally, we clarified that during relatively humid T periods, some areas experience deep cloud development under the influence of continental aerosols, leading to precipitation. This does not conflict with the theory proposed by Ackerman et al. (2004) because they target different subjects. Additionally, some studies, such as Haghighatnasab et al. (2022), have also addressed this point.

**Reference**

Haghighatnasab, M., Kretzschmar, J., Block, K., and Quaas, J.: Impact of Holuhraun volcano aerosols on clouds in cloud-system-resolving simulations, Atmos. Chem. Phys., 22, 8457–8472, https://doi.org/10.5194/acp-22-8457-2022, 2022.

Line 274: "Continental aerosols have a significant impact on precipitation in ECO (Fig. 7). In the absence of continental aerosols, during the more unstable NT with stronger vertical motion, the rainwater path (RWP) in ECO (4.7 g·m⁻²) is much higher than during T (2.2 g·m⁻²). However, in the environment with high aerosol concentrations that includes continental aerosols, Nd increases, and cloud droplet sizes decrease to below the precipitation threshold, leading to precipitation being significantly suppressed in areas with high rainwater content (RWC) in the Sen experiment." These sentences are hard to understand. Are continental aerosols included, or is this the Sen experiment? Which RWP is greater than which other RWP? In these sentences, why not include mention of the titles of the panels in Fig. 7 (Sen_RWC_NT, Sen_RWC_T, etc.)? In general, I don't understand why Sen_RWC_T has *less* rain than Control_RWC_T (see Figs. 7 and 8). It seems counterintuitive, from the perspective of the Albrecht lifetime hypothesis.

We thank the reviewer for highlighting this unclear formulation. We have rewritten this paragraph to improve its readability.

Regarding the comment about whether the fact that Sen_RWC_T is lower than Control_RWC_T under the Albrecht lifetime hypothesis goes against intuition, we believe it does not. Albrechtsuggests that for shallow marine clouds, an increase in aerosol leads to an increase in $N_d$, which results in a reduction in droplet size, thereby suppressing drizzle and increasing the low-level cloud lifetime. In this study, the main precipitation area during NT and the precipitation area in the southeast during T align with this theory. What appears to contradict Albrecht's theory is the increase

in RWC in the northwest precipitation area during T, but we argue that there is no conflict. The increase in RWC in that area is due to clouds developing more deeply with continental aerosols, which enhances condensation and collision processes to form stronger precipitation. This does not satisfy the premise of "low-level" and "shallow marine clouds" (this study focuses on winter liquid-phase clouds, most of which are shallow clouds, but not all. Under the influence of continental aerosols, clouds become deeper and cloud top heights can exceed 4,000 meters in some areas), so there is no contradiction. Additionally, related modeling studies on liquid-phase clouds, such as Haghighatnasab et al. (2022), also point out that "Inside the [polluted airmass], there is a decrease in light rain and an increase in heavy rain … Cloud droplets must grow larger, leading to deeper clouds in order to reach the size that they can start to precipitate. This leads to a shift in the LWP distribution to the higher values inside the plume." This is similar to the situation in this study.

**Reference**

Haghighatnasab, M., Kretzschmar, J., Block, K., and Quaas, J.: Impact of Holuhraun volcano aerosols on clouds in cloud-system-resolving simulations, Atmos. Chem. Phys., 22, 8457–8472, https://doi.org/10.5194/acp-22-8457-2022, 2022.

Line 298: "As shown in Fig. 8, RH exhibits a relatively clear negative correlation with LTS over the four areas during NT and the following 12 hours. In contrast, LTS is relatively high, and there is no significant correlation between RH and LTS during T, suggesting that changes in RH during this period are mainly driven by horizontal variations 300 in temperature and water vapor content."  I don't see the "clear negative correlation with LTS".

We revised this paragraph to discuss the relationship between supersaturation and RH and LTS based on statistical analysis.

Line 318: "Overall, while continental aerosols lead to a decrease in CLWP in some conditions, they generally have a positive impact on CLWP."  Do you mean *the presence of* continental aerosols?

We have rewritten this paragraph to discuss aerosol-cloud interactions in different environments based on a more accurate description and statistical analysis.

Line 321: "The additional continental aerosols also accelerate a transition to ice-phase and mixed-phase clouds by elevating cloud top heights, for example, during the first 12 hours in areas A and B."  Where is ice plotted?  By the way, in Fig. 8, I don't even see a line for CWP.  The legend for CWP is blank.

Previously the CWP was indicated by blue shading. In the revised manuscript, we filtered out the grids containing ice phases to avoid their interference with the analysis of liquid-phase cloud aerosol-cloud interactions, and the CWP is no longer needed and is removed.

Line 324: "1) for low LTS conditions,such as in areas A, B, and C from 12:00 on the 2nd to 00:00 on the 3rd, continental aerosols cause a relatively brief and explosive increase in CLWP. 2) For high LTS conditions, such as during T in the four areas, continental aerosols lead to relatively prolonged

high values of CLWP (around 10 hours) in ECO." I don't see this in Fig. 8. Rather, I see peaks in CLWP both between 12:00 on the 2nd to 00:00 on the 3rd, and also during period T.

We have rewritten this paragraph, supporting the discussion with more accurate statistics.

Line 330: "In terms of precipitation, during the relatively dry NT, continental aerosols cause reductions in both the number concentration of raindrops (Nr) and RWP." The RWP line is barely distinguishable from zero. It is hard to see.

We changed the RWP in Figure 8 to RWP multiplied by three to make the lines more visible.

---

## Author Comment (AC2)

**Response to the Comments of Referees**

**Journal:** Atmospheric Chemistry and Physics
**Manuscript Number:** egusphere-2024-3662
**Title:** Aerosol-cloud interactions in liquid-phase clouds under different meteorological and aerosol backgrounds
**Author(s):** Jianqi Zhao, Xiaoyan Ma, and Johannes Quaas

We thank the reviewers and editor for providing helpful comments to improve the manuscript. We have revised the manuscript according to the comments and suggestions of the referees.

The referee's comments are reproduced (black) along with our replies (blue). All the authors have read the revised manuscript and agreed with the submission in its revised form.

**Anonymous Referee #2**

**Summary**

In this study, the authors utilize WRF case study simulations to evaluate the influence of meteorology and aerosols on cloud properties. Simulations are set up to capture the eastern coast of China and the marine clouds off the coast. Four types of sensitivity cases are evaluated, testing the influence of anthropogenic aerosols (Control vs. Sen, which has anthropogenic aerosols removed) and of aerosol transport from the mainland over the ocean (T vs. no transport, NT). The Control simulation, which assimilates reanalysis information, is compared against satellite and surface observations and reanalysis to establish the skill of the WRF set up. Then the four types of cases are contrasted to evaluate meteorological (Control T vs NT) and aerosol (T vs. NT, Control vs. Sen) influences. Conclusions are drawn about the influence of aerosols in varied meteorological states and the influence of meteorology on the marine cloud systems.

The conceptualization of this study is interesting and the mission, to understand how meteorology and aerosols influence marine clouds, is an important one. However, there are several critical problems that need to be addressed before publication can be considered. Their analysis is not sufficiently robust to support their conclusions at this point. They have not done the necessary statistical testing and quantification of uncertainty needed for identifying aci signals above the noise of the complex system they are examining. The skill of the underlying Control simulations and whether they can reproduce observed aerosol and cloud properties is not robustly demonstrated (Section 3.1), impairing the subsequent sensitivity tests and comparisons conducted in the final sections (3.2-3.3). Their conclusions (Sections 3-4) infer causality from largely correlative and qualitative comparisons that are confusingly presented. While there could be potentially useful insights developed about meteorology-aerosol influence on these marine cloud systems, these analysis flaws impair that. It is also concerning that the simulation startup files and key outputs are not archived publicly for this paper, making their work irreproducible and unable to be evaluated by the community. While the introduction has good awareness of the literature, there is a lack of discussion of the results of this paper and how they relate to the literature. Because I am not

confident as to the skill of the simulations or the robustness of any of the conclusions, as well as the potentially large task to improve the analysis to rectify these issues, I recommend rejection and resubmission. See below for more details.

We thank the reviewer for their support for the overall scope and method of the study, their very good summary, and their constructive remarks. We hope that our thorough revision and additional analysis now improved the study substantially. The details are provided below and in the revised manuscript.

**General Comments**

There is a fatal lack of statistical analysis and uncertainties in this paper. The results appear to be largely qualitative comparisons that are correlative and not causal. Please perform statistical tests on all analysis (e.g., t-tests or non-parametric equivalents to evaluate distributions); establish the bias of the Control simulation against observations and propagate this uncertainty through to the sensitivity simulations; and provide uncertainty measures for all the comparisons you make (e.g., 2 standard error, interquartile range, etc.).

This is a very useful suggestion by the reviewer that helped corroborate the quantitative findings. We now use temporal and spatial correlation coefficients, and normalized mean bias (NMB) to give a statistical assessment of the simulation results.

Following approaches previously applied in the literature, the principle of sensitivity experiments in the detection-and-attribution framework is to compare Control experiments with observations to confirm that the model can reproduce the observations (e.g., numerical, spatial distributions, and temporal variations) within acceptable errors, and then to perform both qualitative and quantitative analyses based on the differences between Control experiments and sensitivity experiments.

For the comparison between the observations, Control, and Sen experiments, we added the 25th to 75th percentile range, mean, median, range with 1.5 Interquartile Range (IQR), and outliers in Figs. S1-S3. For the comparison between the Control and Sen experiments, due to the large sample size (reaching 360,000), the plotting software was unable to process the median, range with 1.5 Interquartile Range (IQR), and outliers. Therefore, we presented the mean and the 25th to 75th percentile range. This addresses the issue of lacking uncertainty measures in our previous manuscript.

Please also be clear about what you are comparing and what domains you are computing variable values (means?) for in your plots. While the red box and a, b, c, d regions are discussed, it is unclear when they are used and what the criteria is for their use. Across this paper, please include uncertainty bars and statistical quantification to demonstrate when differences between NT and T and between Control and Sen are significant (and at what confidence level). Otherwise, the figures are visually engaging and well executed.

In light of this reviewer remark, we have rewritten Section 3.3, revising and providing several new figures for a more organized analysis. We provide uncertainty (the 25th to 75th percentile range, mean, median, range with 1.5 IQR, and outliers) and statistical quantification (temporal and spatial

correlation and normalised mean bias) in the text.

Provide simulation start files and output so that your results can be reproduced and evaluated by the community.

This is a very good suggestion. We have included the simulation start files as Table S3 in the supplement, and the model output has been uploaded to Zenodo (https://doi.org/10.5281/zenodo.15001023).

**Detail Comments:**

Lines 12-24 (Abstract): I don't think you can make these claims without showing that the results are statistically significant (e.g., at 95% confidence). You also need to establish that these are causal linkages, not correlative, which is unclear from the analysis.

In response to this concern, we now provide spatial correlation, NMB, and uncertainty information in the model evaluation. We have revised abstract and section 3.3 to more fully illustrate this by providing statistical information, and by giving statistical figures and analyses of the variation of cloud parameters witha meteorological and aerosol elements.

Figure 1 caption: define what is NT and T.

Corrected.

Section 2.3: It's important that you have matched the satellite observations and simulations so that their sampling time and general retrieval criteria are similar. Very nice.

Thank you. Such a match allows us to evaluate the simulation more rationally.

Line 157-158: How does not including thinner clouds influence your results? Please comment on the potential biases that this could introduce into the results.

Thinner clouds (COT < 5) are excluded only in the evaluation of the model but included in the analysis of the simulation results. This is because MODIS has low retrieval accuracy for thinner clouds, and researchers typically filter out clouds with COT < 5 when using this data. The figure below (Fig. R1) shows the evaluation results without filtering out these clouds. The impact on CLWP is relatively minor, but it greatly affects $N_d$, which is derived from CER and COT. This can lead to extreme $N_d$ values due to uncertainties in satellite retrievals and the inapplicability of the $N_d$ calculation formula under relatively extreme conditions.

[Figure]

Fig R1. Same as Fig. 3, but includes clouds with COT < 5.

Line 172-174: I don't understand why you include the non-liquid clouds as zeroes... if you aren't looking at the total, why do you need to add in this offset?

In Section 3.2, our main concern is the overall effect of continental aerosols on the liquid-phase cloud in ECO, using cloud parameters averaged over the entire ECO region and over the entire time period. Calculating only values within the liquid-phase cloud without setting the values of the no cloud or non-liquid-phase cloud grids to zero does not reasonably represent the overall ECO situation and may affect the representativeness of the average due to individual extremes, thus obscuring the true aerosol-cloud relationship. In addition, there were some problems with our presentation, which appeared to only set the ice and mixed phase cloud grids to 0 and not process the no cloud grids, so we have changed it to "where the values from vertical layers and grid points without clouds or with non-liquid-phase clouds, set to zero, are also included in the calculation".

Figure S1 (and other profile comparisons): Please state whether the values shown in S1 are means, add uncertainties (e.g., 2SE, 25-75%), and indicate where the profiles are being computed over (what is the outer domain, the whole region?).

We added the 25th to 75th percentile range, mean, median, range with 1.5 Interquartile Range (IQR), and outliers in Fig. S1.

We provided the explanation "(all colored areas in Fig. 1, including the parts obscured by the Reference Vector)" at the first occurrence of "outer domain" in the second paragraph of Section 2.1.

Line 189-190: You are suggesting that the Control agrees with the meteorological behavior (please show statistically, see above), which you note it must due to data assimilation. However, that says nothing about how well it does at getting the aerosol (and cloud) properties so remove the "in consequence" statement based on S1.

We have added statistical information based on the above suggestions. Additionally, the original phrase "as expected thanks to the data assimilation" was too absolute and overlooked the accuracy of the reanalysis data as well as the contribution of the model's high performance. We have changed it to "supported by data assimilation."

Fig. S1 does not directly assess the model's ability to simulate aerosols and clouds, but the reasonably reproduced meteorological fields serve as the foundation for simulating aerosols and clouds. The phrase "in consequence" was indeed somewhat inappropriate, so we changed it to "Based on the reasonably reproduced meteorological fields".

Figure 2 and 3: As far as I can tell, these are qualitative comparisons. It does look promising but it is hard to tell from this what the actual skill of the Control simulation is in aerosol and cloud properties (which I agree is crucial to establish in this section). I recommend that you focus on the Control to observation comparison in this section and remove the Sen column (see comment below). Instead, I strongly encourage you to make the last column in both these figures the difference map between Control and observations so that it is obvious what the differences are and how they may geographically vary. You can then use this to compute the bias between the Control and observations for these various parameters, which you can then propagate through the other comparisons later (i.e., to show that the meteorology or aerosol influence signal is larger than the bias in the Control

simulations). I also strongly suggest that you do a statistical comparison between the observations and Control. For example, you could look at a coarser grid and test whether the value distribution for each coarse grid box is statistically similar (e.g., with a t test, if assuming they have a normal distribution, or a non-parametric equivalent) and what their r values are at 95% confidence. For Figure 3 (and Figure S2) please also include the land borders so easier to compare across figures.

Following the reviewer suggestion, we removed the Sen experiment simulation and replaced it with the difference between the simulations of the Control experiment and the observations.

Based on your comment and relevant studies, we statistically compare simulations and observations using four metrics: the differences (show the magnitude and distribution of discrepancies), the normalized mean bias (quantify the differences), uncertainties (in Figs. S1-S3, -include the 25th to 75th percentile range, mean, median, range with 1.5 IQR, and outliers) and the spatial correlation coefficient (the Pearson product-moment coefficient, which measures the spatial relationship between simulation and observation).

The land boundary is not included in Fig. 3 and Fig. S2 because the entire region is offshore (the inner domain), this is now clarified in the Caption. Their locations are shown in the red boxes in Figs. 1 and 2. We note in the last paragraph of Sections 2.3 and 3.1 that the evaluation of the cloud simulations and the follow-up model-based analyses are based on the results of the inner domain simulations.

Line 203-212, Section 3.1: I would recommend not discussing or introducing the Sen experiment until the next section. It's clearer to really focus on establishing that Control is reliable by comparing to observations in this section. It's confusing to be discussing Sen at the same time especially as its unclear at this point whether we can trust the underlying simulation. It would also be helpful to introduce more details of how you have done the Sen experiment (minimal in the methods) before discussing the Sen results. Specifically for this paragraph, you need to do a statistical comparison between the Control and Obs (to show the Control captures the key cloud properties) and the Control and Sen (to establish the magnitude of the difference and whether that signal is larger than the Control to Obs bias). Visually, the cloud properties look much more different from the Obs than the aerosol. This likely impairs the ability to evaluate cloud changes in the Sen experiments.

the reviewer has a good point here that the evaluation of the simulation results in section 3.1 should indeed focus primarily on the differences between Control experiment and observation, and the Sen experiment, which is counterfactual, causes some confusion when placed here. So we have removed the Sen experiment and evaluated the Control simulation more adequately based on statistical comparisons that were added as you suggest above. The Sen experiment now only appears in the subsequent simulation-based aerosol-cloud analysis, which analyzes the impact of continental aerosols by comparing the Control and Sen experiments, based on the simulation confidence established by evaluating the Control simulation.

We added the description of the Sen experiment in the last paragraph of Section 2.1 '… continental

aerosol (anthropogenic, dust, and bioaerosols, as well as aerosol and chemical initial and boundary conditions from CAM-chem) emissions are turned off'', and provided the simulation start file for the Sen experiment in supplement Table S3

The difference between simulated and observed cloud parameters is indeed significantly higher than that of aerosols, which is due to a number of reasons, including the difficulty of cloud simulation, errors in the driving data and errors in satellite retrievals. However, the main reason is the difficulty of cloud simulation, and the related simulation studies have admitted that the existing computational ability and theoretical level cannot realize a perfect simulation of clouds. Most evaluations of cloud simulations use, for example, spatial distribution plots or histograms of frequency distributions to compare the differences between simulations and observations, without calculating more statistical information. The simulation is considered reliable as long as the values and distributions of the observed cloud parameters can be reasonably reproduced within acceptable errors.

Line 219-220: I disagree with this statement. Please include more extensive evaluation including statistical assessment of how well the model can reproduce the meteorology, aerosol, and cloud parameters to establish the skill of the Control simulation. Because you are assimilating the meteorological information, that's reasonable to get right. However, you need to demonstrate statistical skill for the aerosol and cloud parameters to analyze aerosol and cloud later in the sensitivity experiments. Please separate out the Sen experiment discussion until after you have shown that the Control experiment has sufficient skill. It will help clarify this story and your conclusions a lot.

We provided a more comprehensive evaluation by adding statistical assessment, as the reviewer suggested above. In addition, we removed the Sen experiment from Section 3.1 and discuss the Sen experiment in subsequent analyses after proof of the confidence of Control simulations.

Section 3.2, 3.3, Conclusions: Unfortunately, without establishing the bias in the Control relative to the observations, it is unclear whether any of the sensitivity test comparisons are statistically significant and thus informative.

As stated above, we supplemented the evaluation of the Control experiment by adding statistical information to provide confidence in the subsequent analyses.

Figures 4-6: For all the profile and distribution (mean?) comparisons, please indicate what part of the domain they are over (the red box?). Please add some uncertainty measure, such as 2SE or 25-75% for these to show the statistically significant differences between simulations.

They are means, and we add this description in the captions of Figs. 4-6. We add the domain note ("In the subsequent sections (3.2 and 3.3), we analyze aerosol-cloud interactions using high-resolution inner-domain (the area in the red box as shown in Figs. 1 and 2) simulations") at the beginning of the analysis section (last paragraph of Section 3.1).

We have added shading in these figures to indicate the 25th to 75th percentile range.

Line 223: To test this, please establish that the Control simulation sufficiently captures these behaviors first for both T and NT cases.

We have supplemented section 3.1 with quantitative evaluations to confirm the performance of the Control simulation as you suggested.

Line 226-227: From the Control or observations? Is this all the region or just the red box collapsed into profiles?

The analyses in Sections 3.2 and 3.3 are both based on inner domain (in the red box) simulations, which we noted in the last paragraph of Section 2.3 "The analysis in this study is based on high-resolution simulations in the inner domain ..." . It may be easily omitted, so we added "In the subsequent sections (3.2 and 3.3), we analyze aerosol-cloud interactions using high-resolution inner-domain (the area in the red box as shown in Figs. 1 and 2) simulations" at the end of section 3.1.

Line 233-235: Where is this shown?

It is shown in Fig. 5a and d (aerosol spectral distribution) for the Control and Sen experiments in comparison. We added the statement to this sentence.

Line 243-244: Do you show that these are causal linkages? Or are these correlations for the behaviors and cloud properties during NT and T cases? If the latter, they should not be discussed as causal. Please be clear of your comparisons and whether they are causally informative here and throughout.

It is the latter, and we have modified it to "The difference between the two periods lies in the fact that during NT, when the atmosphere is relatively unstable, the cloud height is higher than during T. Meanwhile, during T, which has higher water vapor and aerosol contents, Nd and CLWP are higher than during NT. Meanwhile, during T, which has higher water vapor and aerosol contents, $N_d$ and CLWP are higher than during NT", avoiding the unrigorous use of causal descriptions.

Line 244-246: Please expand on what you mean here, I don't understand the point you are making.

We added a more clear description "(i.e., aerosols entering the cloud can be activated regardless of particle size, rather than large particles being fully activated and only a small fraction of small particles being activated)".

Line 246-248: It's not clear this is causal to me, are you gathering this from correlation or comparisons between Control and Sen?

This is illustrated by a comparison of Control and Sen. As described in the previous sentence, in the clean case with only sea-salt aerosols (Sen), the aerosols in each bin are largely fully activated. Then more ocean emissions during T lead to more sea salt aerosols in the atmosphere, which in turn leads to more cloud droplets.

Line 250: Can you show that "similar to that near the source area."?

We added the aerosol number concentration profiles of ECO and its aerosol source area as Fig. S4.

Line 253: You discuss this as causal ("led"), is this the comparison between Control and Sen? Otherwise, correlative, please change this language and clarify.

We avoided the interference of radiative effects in the analysis of aerosol-cloud interactions by turning off the radiative feedback of aerosols and clouds. The statistics in Fig.S1 and Table S2 also indicate that the Control and Sen experiments run under similar meteorological conditions, with the only difference being the presence of continental aerosols. The differences in cloud parameters between the two experiments are also caused by continental aerosols, making the use of "led" appropriate in this context.

Line 259-260: Please show the supersaturation to support this statement.

We added the supersaturation profiles in Fig. 5.

Line 260-262: Can you please rephrase? I'm not sure I understand what you mean here.

In light of the reviewer remark, and also by further consideration by which we found this to be insufficient to support us in drawing further conclusions, and its presence or absence does not affect the main content of our manuscript, we have deleted this paragraph.

Line 262-265: Are these statistically significant effects?

The uncertainty information added in Figs. 4-6, along with the qualitative and quantitative comparative analysis of the parameters from the two experiments, supports this conclusion. No additional statistical tests were conducted here. To avoid confusion, we replaced 'significantly' with 'greatly'.

Line 274: At what level is this statistically "significant"?

What we want to express here is the strong influence of aerosols on precipitation as shown in the figure. This is shown by the qualitative and quantitative comparison of the means rather than by a significance test. The use of "significant" may be misleading in terms of statistical significance, and we have changed it to "strong".

Line 280: Which figure shows that it's unstable and dry? Please indicate.

It is shown in Fig. 4, and we added the note here. Clearly higher vertical temperature lapse rate and lower water vapor content during NT support "The relatively unstable and dry environment during NT".

Figure 7: This difference plot is very helpful. Please do something like this for all the other map comparisons (Control vs. Obs or Control vs. Sen). However, is this difference larger than the bias between Control and Observations? Please have some indication on here for what is a statistically significant difference. Can you compare the control and sen distributions for the levels in the gridded regions, as suggested before, to determine this?

We have added the difference plot to the other map comparisons (Figs. 2 and 3).

We added the 25th to 75th percentile range, mean, median, range with 1.5 IQR, and outliers in Figs. S1-S3. The comparisons of aerosol and cloud parameters in the figures indicate that the difference between Control vs. Sen is larger than the difference between Control vs. Obs. The uncertainty added in Fig. 6d provides statistical information that supplements the precipitation differences between the two experiments and two time periods.

Line 295-296: I don't understand the rationale here. Usually, you want to look at how the clouds are influenced by the environment, so you pick either a sub/surface level (925hPa, Wood 2012) or something above the cloud in the free troposphere (e.g., 700hPa) (e.g., Klein et al. 2017). Can you show that this metric is giving you useful information about how the cloud is being influenced by the environment?

In the relevant studies on liquid-phase clouds that we have reviewed, LTS and RH are widely used as indicators to characterize the impact of the environment on clouds. LTS is calculated as the difference between the potential temperature at 700 hPa and at the surface. Our analysis in Section 3.2 shows that in the cases studied, the vast majority of cloud water appears at altitudes ranging from the surface to 1300 m. Therefore, using the vertically weighted average RH from the surface to 1300 m most intuitively and directly reflects the impact of environment elements on aerosol-cloud interactions by influencing aerosol activation, cloud droplet growth, and related physical processes. In addition, we have added vertical weighted supersaturation from the surface to 1300 m in this section.

Figure 8: This is a very hard figure to read... I suggest separating out more of these parameters so you can tell a clearer story. Also, if you are discussing correlations over time it would be informative to use lagged correlations (showing r when at some confidence level, e.g., 95%). These could help to test hypotheses of what is influencing the cloud parameters and which factor is comparatively more important. I noticed that the regions you are comparing (a, b, c, d) is marked in Figure S2. Please also mark them on the maps in the main text so it is clear. Also, please explain the rationale behind choosing these boxes as they seem widely dispersed and quite small. Why are these informative? Is there a flow between the boxes you are trying to capture?

In response to this reviewer comment, we removed some parameters from Figure 8 and added three new figures to analyze the aerosol-cloud interactions more clearly.

For the parameters in Figure 8 that exhibit a certain lag, we applied lag correlation analysis (LTS and supersaturation). For other parameters that show synchronized changes in the figure, we directly used time correlation analysis.

We added markers for these regions in Figure 3a. Our goal is to analyze the time variations of relevant parameters in regions where aerosol and cloud parameters exhibit different changes due to continental aerosol influence, in order to make some hypotheses. To avoid averaging obscuring specific information, we have added statistical analysis for these hypotheses based on all regions and

times in Section 3.3.

Line 303-305: Please clarify what you mean here.

We removed this sentence and provided a more comprehensive description of the relationship between $N_d$ and $N_a$ by adding Fig. 9 and its analysis.

Line 305-307: Can you show this? Do you see that the critical supersaturation is modified in the updrafts?

We removed this sentence and added a statistical analysis of the relationship between supersaturation and LTS and RH.

Line 307-308: You say "(due to the transient and localized nature of supersaturation, RH is used to represent the overall supersaturation intensity in this environment)". If this is how you are using this integrated RH quantity, you need to explain this at the beginning of the controlling factor discussion and demonstrate that it is really containing the supersaturation information you believe it is. Because you are integrating over the cloud as well (not just the sub-cloud layer) and weighting to where there is more moisture, I suspect that this is telling you about how juicy the cloud is, not about the moisture in the updraft that the aerosol is experiencing when lofted and activated.

We removed this paragraph and incorporated supersaturation into the analysis in Section 3.3 to examine the relationship between LTS, RH, and supersaturation, as well as the relationship between supersaturation and aerosol-cloud interactions.

Line 309: As noted above, if this is essentially the cloud liquid measure it likely does not inform you of the impact on aerosol activation. I suggest either looking at RH below cloud or, preferably, pull the critical supersaturations in the updrafts so you can get the activation. Please demonstrate that the RH metric has some relationship with the activation potential you are claiming here. What about the differences in aerosol composition and size that could also impact activation capability between the anthropogenic dominated Control and sea spray dominated Sen? How does that influence aerosol activation ability?

We now incorporated supersaturation into the analysis in Section 3.3.

The impact of aerosol composition and size on activation is indeed one of the reasons for the differences between Control and Sen. We have explained this in Section 3.2 ("…the environment (Fig. 5c) could not meet the high supersaturation requirements for fully activating such a large number of small, low-hygroscopic aerosols").

Line 327-329: I don't see how you can disentangle the meteorology and aci effects to make this statement. Please show the work that supports this. One way to isolate the aci from the meteorology is if you show that the meteorology is the same in the Control and Sen experiments for the NT and T cases, separately. If you were able to establish that they are relatively similar (for the respective NT and T cases), then you may have some causal connection in the difference between Control and Sen that is informative of aci. Could you quantify the contributions of adjustments and Twomey effect

using something like the Erfani et al. 2022 method? You could maybe then infer something from comparing the NT and T aci contributions. Please include uncertainties and statistics to show if the differences are significant.

We analyzed the uncertainty, temporal and spatial correlations, as well as the NMB, of various meteorological factors in the Control and Sen experiments. As shown in Fig.S1 and Table S2, the uncertainty information for Control and Sen is essentially consistent, the correlation coefficients for each factor between the two experiments range from 0.99 to 1, and the NMB is within ±0.3%, indicating that the meteorological backgrounds of the two experiments are similar.

Erfani et al. (2022) decomposed the radiative forcing of adjustments and the Twomey effect, while this study focuses on the impact of aerosols on clouds under different environments. To isolate the aerosol-cloud interaction signals, the radiative effects of aerosols and clouds were turned off. We examined the influence of $N_a$ on $N_d$ and CER in different environments, as well as the resulting changes in CLWP, RWP, and cloud lifetime. The former corresponds to the Twomey effect, while the latter represents rapid adjustments.

Line 330: How do you know they are causing this precipitation change?

As mentioned above, we have statistically demonstrated that Control and Sen are in similar meteorological fields. The only difference in the setup of the two experiments is whether or not continental aerosols are included. Therefore, it can be said that the difference in precipitation between the two experiments is caused by continental aerosols.

Additionally, in the revised manuscript, we have rewritten Section 3.3 to provide a more statistical explanation of the impact of aerosols on precipitation.

Line 340-342: Your analysis does not support this conclusion. See previous points about i) statistical testing, ii) biases in Control from obs being larger than Control-Sen differences, iii) decomposing Twomey and adjustment effects, and iv) controlling for meteorology. For the latter, you likely need to show that the meteorology is relatively unchanged between Control and Sen except for their cloud and aerosol, otherwise you can't distinguish the meteorology and aerosol influences.

i) We provided statistical tests in Section 3.1.

ii) We demonstrated this by adding uncertainty information in Figs. S1-S3.

iii) In Section 3.3, we comprehensively discussed how $N_d$ and CER change with $N_a$ in different environments, as well as the subsequent changes in CLWP, RWP, and cloud lifetime. The former reflects the Twomey effect, while the latter represents rapid adjustments.

iv) As shown in Fig. S1 and Table S2, we provided statistical evidence for the similarity of the meteorological fields in the Control and Sen experiments.

Line 352-353: I disagree with this. Statistical analysis needs to be employed here to establish Control captures real world behaviors and the bias from observations. This bias will be essential to account

for in your subsequent analysis.

We now supplemented Section 3.1 with statistical analysis to demonstrate the reliability of the simulations.

Line 355: How are the aerosols disabled? Does this effect anything else about how aerosols are handled or just the number?

We disabled the emissions of continental aerosols in the model setup and removed continental source gases and aerosols from the model's aerosol and gas initial and boundary information. This will affect the aerosol composition, number, and their associated physical and chemical processes. Since we disabled aerosol and cloud radiative effects during the re-simulation, the impact on aerosol-cloud interactions, under the condition that the meteorological conditions of the two experiments are essentially consistent, is limited to the quantity and composition of aerosols.

Line 355-357: These four different comparisons need to be discussed more clearly throughout so that it's obvious when you are contrasting NT and T (not causal) and Control and Sen (theoretically more causal if you are only changing the aerosol and the meteorology is the same). I found this quite confusing in reading the text and evaluating the plots (which are also over varying domains).

We made substantial revisions to Sections 3.2, 3.3 and 4, providing more figures and statistical information to make the analysis more comprehensive and clear.

Line 359-360: Where do you show this: "the atmosphere fails to enable full activation of aerosols during both periods"?

This can be seen through the aerosol size distribution in the Sen and Control experiments (Fig. 5a), where the activation ratio of small aerosol particles in the Control experiment is clearly lower than that in the Sen experiment.

In addition, we have rewritten our conclusions based on the new results, and in this paragraph we synthesize the relationship between environment, supersaturation, aerosols, and clouds.

Line 365-367: This does not seem like a surprising conclusion, reference previous literature?

We rewrote Section 4, and this content no longer appears in the conclusion.

Line 369: Please show the representative areas on the maps in the main figure and make it clear the criteria for how you chose them and why they are representative.

We marked their locations in Fig. 3a and explained our selection method and their representativeness in the first paragraph of Section 3.3.

Line 371: Where do you show this: "aerosols only affect clouds during their development stage without noticeably impacting their lifetime"?

We added Fig. 11 to illustrate the frequency of cloud occurrence, as well as the variations of CLWP and RWP with RH and LTS, along with the differences between the two experiments. The figure

shows that there is no difference in the location and frequency of cloud occurrence between the Control and Sen experiments (at least from the perspective of the model output's 1-hour temporal resolution).

Line 373: Please look at the supersaturation itself to support this: "by supersaturation, with low supersaturation limiting the full activation of continental aerosols."

We added supersaturation and the corresponding analysis in Sections 3.2, 3.3 and 4.

Line 375-376: Do you show this change in phase or are you inferring it? Please show work or cite literature.

In the previous version, CWP greater than CLWP in Fig. 8 indicated the presence of non-liquid phase clouds. In the revised manuscript, we focused on aerosol-cloud interactions within liquid phase clouds, removed these cases, and only analyzed liquid phase cloud samples.

Line 379-381: Do you show this somewhere? Please clarify what you mean here and indicate the supporting work.

This can be seen from the comparison of $N_d$ changes with and without precipitation in the previous Fig. 8. In the revised manuscript, we conducted a more comprehensive analysis of the impact of precipitation using the statistics in Fig. 9.

Line 385: Please provide all the necessary setup files and the key outputs from your simulations at an archive (e.g., like the free Zenodo) so that your simulations can be reproduced by the community and evaluated.

We have included the model setup files as Table S3 in the supplement, and the model output has been uploaded to Zenodo (https://doi.org/10.5281/zenodo.15001023).

Section 4 (or before): Please include some discussion of how your results relate to prior work in the literature and provide some contextualization of this study.

We provided this content in the first and last paragraphs of Section 4.

**References**

Erfani, E., Blossey, P., Wood, R., Mohrmann, J., Doherty, S.J., Wyant, M., O, K., 2022. Simulating Aerosol Lifecycle Impacts on the Subtropical Stratocumulus-to-Cumulus Transition Using Large-Eddy Simulations. JGR Atmospheres 127, e2022JD037258. https://doi.org/10.1029/2022JD037258

Wood, R., 2012. Stratocumulus Clouds. Mon. Weather Rev. 140, 2373–2423. https://doi.org/10.1175/mwr-d-11-00121.1

Klein, S.A., Hall, A., Norris, J.R., Pincus, R., 2017. Low-Cloud Feedbacks from Cloud-Controlling

Factors: A Review. Surveys in Geophysics 38, 1307–1329. https://doi.org/10.1007/s10712-017-9433-3